# Inverse association between total bilirubin and type 2 diabetes in U.S. South Asian males but not females

Aayush Visaria[1]*, Alka Kanaya[2], Soko Setoguchi[1], Meghana Gadgil[2], Jaya Satagopan[3]

1 Center for Pharmacoepidemiology and Treatment Sciences, Rutgers Institute for Health, New Brunswick, New Jersey, United States of America, 2 Department of Medicine, University of California–San Francisco, San Francisco, California, United States of America, 3 Department of Biostatistics and Epidemiology, Rutgers School of Public Health, Piscataway, New Jersey, United States of America

* aayush.visaria@rutgers.edu

## Abstract

### Aims

United States South Asians constitute a fast-growing ethnic group with high prevalence of type 2 diabetes (T2D) despite lower mean BMI and other traditional risk factors compared to other races/ethnicities. Bilirubin has gained attention as a potential antioxidant, cardio-protective marker. Hence we sought to determine whether total bilirubin was associated with prevalent and incident T2D in U.S. South Asians.

### Methods

We conducted a cross-sectional and prospective analysis of the Mediators of Atherosclerosis in South Asians Living in America (MASALA) study. Total bilirubin was categorized into gender-specific quartiles (Men: <0.6, 0.6, 0.7–0.8, >0.8; Women: <0.5, 0.5, 0.6, >0.6 mg/dl). We estimated odds of type 2 diabetes as well as other cardiovascular (CV) risk factors using multivariable logistic regression.

### Results

Among a total 1,149 participants (48% female, mean [SD] age of 57 [9] years), 38% had metabolic syndrome and 24% had T2D. Men and women in the lowest bilirubin quartile had 0.55% and 0.17% higher HbA1c than the highest quartile. Men, but not women, in the lowest bilirubin quartile had higher odds of T2D compared to the highest quartile (aOR [95% CI]; Men: 3.00 [1.72,5.23], Women: 1.15 [0.57,2.31]). There was no association between bilirubin and other CV risk factors.

### Conclusion

Total bilirubin was inversely associated with T2D in SA men but not women. Longitudinal studies are needed to understand temporality of association.

**Data Availability Statement:** The data underlying the results presented in the study are available from MASALA investigators at University of

California San Francisco upon reasonable request:
https://www.masalastudy.org/for-researchers.
Data are owned and collected by a third party
(MASALA Data Coordinating Center at University of
California San Francisco). A proposal must be
submitted and accepted by the MASALA review
committee before access to the data is granted
(upon signing a data use agreement). More
information regarding data access can be obtained
by contacting ann.chang@ucsf.edu.

**Funding:** The author(s) received no specific
funding for this work.

**Competing interests:** The authors have declared
that no competing interests exist.

## 1. Introduction

Type 2 diabetes (T2D) remains a public health challenge, with prevalence increasing from
9.5% in 1999 to more than 13% ($\sim$34 million) among U.S. adults in 2018 [1]. South Asians
(SA), individuals with origins in Bangladesh, Bhutan, India, the Maldives, Nepal, Pakistan, and
Sri Lanka, in the U.S. have particularly high prevalence of T2D [2] and risk of downstream cardiovascular disease (CVD) despite lower mean BMI and prevalence of other conventional risk
factors [3]. This increased risk of T2D and CVD is likely multifactorial, although causes are
not well understood and are understudied.

In recent years, serum bilirubin has arisen as a potential protective factor for T2D and
related CV factors. Traditionally, bilirubin, a byproduct of hemoglobin degradation, was
known as a cytotoxic waste product in liver disease and hemolytic anemias [4, 5]. However,
stemming from observations of low prevalence of cardiometabolic conditions in patients with
Gilbert's Syndrome, a benign, genetic disorder characterized by elevated unconjugated bilirubin, bilirubin has gained attention as an endogenous antioxidant, cardio-protective marker.
Several observational studies have since demonstrated an inverse association between total bilirubin and cardiovascular disease [6–9] & associated risk factors such as T2D, insulin resistance and dyslipidemia [10–14].

Prior studies in Chinese, Japanese, and Korean populations [7, 15–21] have demonstrated
inverse associations between total bilirubin and cardiometabolic risk factors, but no studies to
our knowledge have assessed the association between total bilirubin and T2D in the fast-growing U.S. SA population. Given the high prevalence of unfavorable lipid profiles and visceral
adiposity in U.S. SAs, both of which can affect or are driven by the liver [22], we hypothesized
that bilirubin, also produced in the liver, may be associated with T2D in SAs.

Hence, the objective of this study was to determine whether total bilirubin is independently,
associated with both prevalent and incident T2D among a cohort of US South Asians.

## 2. Subjects, materials and methods

### 2.1. MASALA study

The Mediators of Atherosclerosis in South Asians Living in America (MASALA) study [23] is
an ongoing, community-based cohort study of SA men and women aged 40–84 without pre-existing CVD from two clinical sites (San Francisco Bay Area at the University of California,
San Francisco (UCSF) and the greater Chicago area at Northwestern University (NWU)). The
baseline examinations used in this secondary analysis were conducted from 2010–2013 (Exam
1, n = 906) and 2017–2018 (Exam 1A, n = 258).

The MASALA study methods and recruitment can be found elsewhere [23]. Briefly, study
participants were included if they had (1) South Asian ancestry defined by having at least 3
grandparents born in one of the following countries: India, Pakistan, Bangladesh, Nepal, or Sri
Lanka; and (2) ability to speak and/or read English, Hindi, or Urdu. Study participants were
excluded if they had physician diagnosed heart attack, stroke or transient ischemic attack,
heart failure, angina, use of nitroglycerin, and/or history of cardiovascular procedures or any
surgery on the heart or arteries. This study was approved by University of California San Francisco's Institutional Review Board, MASALA Data Coordinating Center and Rutgers's Institutional Review Board. This was a secondary analysis of fully de-identified data. Accordingly,
there was no contact with participants. Informed signed consent was acquired during initial
data collection by MASALA investigators.

## 2.2. Study population

Among the original 1,164 participants included in Exam 1 and 1A, we excluded adults with a total bilirubin ≥2 mg/dl (n = 15). Limiting total bilirubin to <2 mg/dl allowed us to examine a general population without potential Gilbert's syndrome. We also excluded participants with missing data on bilirubin and diabetes status (n = 16), leading to a final sample of 1,149 participants.

## 2.3. Exposure: Total bilirubin

Total bilirubin was measured by Quest Diagnostics by spectrophotometry using a serum sample with measures to the nearest tenth mg/dl. Total bilirubin was categorized into gender-specific quartiles (Men: <0.6, 0.6, 0.7–0.8, >0.8; Women: <0.5, 0.5, 0.6, >0.6 mg/dl) as men and women have different distributions of bilirubin and there exists no pre-defined clinical thresholds for total bilirubin. As sensitivity analyses, we also analyzed total bilirubin divided into tertiles.

## 2.4. Primary and secondary outcomes

The primary outcome was T2D, defined by laboratory criteria (fasting plasma glucose ≥126 mg/dl or 2-hour post-challenge glucose ≥200 mg/dL or hemoglobin A1c ≥6.5%), or use of any diabetes medications. The same definitions were used for both prevalent and incident T2D.

We explored several secondary outcomes. These included the 10-year ASCVD risk score greater than or equal to 7.5% via AHA Pooled Cohort Equation [24], presence of any coronary artery calcium (CAC), distal common carotid artery intima-media thickness [IMT], internal carotid artery IMT, presence of CT-based fatty liver, Homeostatic Model Assessment for Insulin Resistance (HOMA-IR) [25], and CV risk factors (HDL-cholesterol, LDL-cholesterol, triglycerides, hypertension, and obesity). CAC scores were determined using non-contrast cardiac CT scans, performed using a cardiac-gated computed tomography scanners (UCSF: Phillips 16D scanner or a Toshiba MSD Aquilion 64; and at NWU: Siemens Sensation Cardiac 64 Scanner (Siemens Medical Solutions, Malvern, PA). All scans were read using standard protocols. Coronary artery calcium Agatston scores were reported for each of the four major coronary arteries and the summed score was used [26]. CAC was divided dichotomously into presence of any CAC and no CAC, as well as into 3 categories: 0, 1–399, and ≥400 to capture clinically significant atherosclerosis. Carotid IMT was determined using high-resolution B-mode ultrasonography. Complete details of the protocol have been published [23] and vascular technicians at both study sites were trained and certified on the scanning protocol by the reading center. Carotid IMT was dichotomized into ≥1.5mm, <1.5mm for the distal common carotid artery and ≥1.0mm, <1.0mm for the internal carotid artery [27]. Fatty liver was defined as a liver attenuation on CT less than 40 Hounsfield units [28].

Dyslipidemia was defined as use of an HMG-coA reductase inhibitor, fibrate, or niacin, HDL < 50 mg/dl if female and <40 mg/dl if male, or LDL ≥160 mg/dl if no diabetes or ≥100 mg/dl for a diabetic participant [29]. High triglyceride levels were defined as ≥150 mg/dl and <150 mg/dl. High blood pressure was defined per the NCEP criterion for metabolic syndrome as a mean SBP≥130 or DBP ≥85 mmHg or current use of antihypertensive medications. Seated resting blood pressure was measured three times using an automated blood pressure monitor (V100 Vital sign monitor, GE Medical Systems, Fairfield, CT) and the average of the last two readings used for analysis. Obesity was defined as a body mass index (BMI) greater than or equal to 27.5 kg/m$^2$ per the World Health Organization's Asian-specific cutoffs [30].

### 2.5. Covariates

We assessed several covariates, including demographic factors (age, education level [<Bachelor's degree, Bachelor's degree only, >Bachelor's degree], family income quartile [<$40k; $40-75k; $75-100k; >$100k], percent of life in the U.S. [<40%, 40–60%, >60%]), socio-behavioral factors (alcohol consumption [ever drinker/never drinker], smoking status [ever smoking, never smoking], total caloric intake (kcal/day), leisure-time physical activity (MET-mins/week), and CV risk factors (HDL-cholesterol, LDL-cholesterol, triglycerides, hypertension, waist circumference, and BMI-based obesity).

### 2.6. Statistical analysis

We first examined baseline characteristics by bilirubin quartiles in men and women separately, reporting continuous variables as mean [SD] or median [inter-quartile range] and categorical variables as N (%). We also illustrated the distributions of total bilirubin by sex. We estimated the odds of T2D using multivariable logistic regression, stratifying by sex. We adjusted for potential confounders selected a priori, including age, BMI, exercise, alcohol consumption, education, income quartile, percent time in the U.S., and metabolic syndrome criteria per NCEP's Adult Treatment Panel (ATP) III (hemoglobin A1c%, systolic BP, HDL, triglycerides, except for outcome of interest), and LDL-cholesterol. We additionally adjusted for smoking status in males but not females due to a very low sample of female smokers.

We conducted several sensitivity analyses in order to assess the robustness of our findings: (1) we performed linear regression analyses among continuous versions of our primary and secondary outcomes, verifying approximate normality of continuous outcomes (HbA1c%, HDL, LDL, triglycerides, systolic BP, diastolic BP, BMI, waist circumference, liver fat attenuation, carotid IMT); (2) identical analyses among 1,093 individuals without elevated liver enzymes, or history of liver disease (aspartate aminotransferase [AST] >40 IU/L or alanine aminotransferase [ALT] >56 IU/L (N = 16 excluded), history of liver disease (N = 40 excluded)). This was done to exclude acute and chronic elevations due to liver disease or inflammation. We also further restricted analysis to participants without CT-based fatty liver (N = 157 excluded); (3) using tertiles to classify total bilirubin (Male: <0.6, 0.6–0.7, >0.7; Female: <0.4, 0.4–0.5, >0.5 mg/dl); (4) performing analysis on Exam 1 participants only using bilirubin tertiles and additionally including caloric intake as a confounder in the multivariable logistic regression model; (5) performing analysis in a subset of individuals not using non-metformin medications for diabetes. We did not treat bilirubin as a continuous variable for sensitivity analyses due to its non-linear association across quartiles and its discrete values due to limited precision of the laboratory estimates to the nearest 0.1 value. Multiple comparisons were adjusted for using Bonferroni correction. All analysis was done with two-sided statistical tests, using SAS 9.4 (Cary, NC) with an alpha level of 0.05.

Lastly, among 749 participants who were seen in Exam 1, follow-up exams were completed from September 2015-March 2018, providing us the opportunity to study incident diabetes among Exam 1 participants without diabetes. Given the low number of incident diabetes cases (N = 49), this was primarily an exploratory analysis. We compared the crude proportions of incident diabetes by bilirubin tertile (Male: <0.6, 0.6–0.7, >0.7; Female: <0.4, 0.4–0.5, >0.5 mg/dl) using Fisher's Exact test.

## 3. Results

Among 1,149 (598 men, 551 women) MASALA study participants, the mean [SD] age was 57 [9] years, 66% had at least a Bachelor's degree, 61% had a household income >$100,000, and 8% lived in the U.S. less than 20% of their lives. Additionally, 38% had metabolic syndrome,

24% had T2D, 47% had presence of any coronary artery calcium, and 23% had 10-year ASCVD risk score ≥7.5% (Tables 1 and 2).

Men and women in the lowest bilirubin quartile had, on average, 0.55 and 0.17 higher HbA1c %, 2.6 and 2.5 mg/dl lower HDL-cholesterol, and higher median HOMA-IR than men and women, respectively, compared to the highest quartile (Tables 1 and 2).

Men, but not women, in the lowest bilirubin quartile had higher adjusted odds of T2D compared to the highest quartile (aOR [95% CI]; Men: 2.93 [1.68, 5.11], Women: 1.20 [0.61, 2.381]; Table 3). Even among patients without liver dysfunction, defined using elevated liver enzymes, self-reported history of liver disease, and CT-based fatty liver, results persisted (Tables 4 and S1-S4 in S1 File).

There was no significant association between bilirubin and other CV risk factors in men or women (Table 5).

### 3.1. Incident diabetes exploration

Among 573 participants without diabetes from Exam 1 with follow-up exam data, 49 (8.5%; 28 men, 21 women) developed T2D. In women, the proportion of incident diabetes by bilirubin tertile was 11.1% in the lowest tertile, 9.0% in middle tertile, 4.0% in the highest tertile (p = 0.22). Among men, the proportion of incident diabetes by bilirubin tertile was 6.5% in the lowest tertile, 8.6% in middle tertile, 12% in the highest tertile (p = 0.38, Table 6).

### 3.2. Secondary outcomes

Among participants with coronary artery calcium and carotid IMT measurements, there was no significant association between bilirubin levels and subclinical atherosclerosis measures (Table 5). Among a subset of participants using metformin for diabetes, there was a similar, but accentuated, association between bilirubin quartile and diabetes in men (aOR [95% CI]; Q1 vs. Q4; Men: 3.30 [1.69, 6.44], Women: 1.00 [0.44, 2.28]).

### 3.3. Additional sensitivity analyses

To test the robustness of the association between total bilirubin level and diabetes status, we recategorized bilirubin into tertiles, examined associations in Exam 1 separately, and assessed associations in the cohort excluding liver dysfunction, defined using elevated liver enzymes, self-reported history of liver disease, and CT-based fatty liver. Results persisted when restricting the cohort to individuals without liver dysfunction (Tables S1-S4 and 4 in S1 File), and when using bilirubin tertiles (Table 3) instead of quartiles. These associations were non-linear, but with a significant trend in odds of diabetes with increasing bilirubin among men (men: p-trend = 0.003, women: p = 0.34). Additionally, men in the lowest bilirubin quartile had 0.51% higher HbA1c compared to those in the highest bilirubin quartile (S5 Table in S1 File).

## 4. Discussion

In this analysis of the U.S. MASALA cohort of South Asian individuals aged 40–84 without known CVD, we found that men, but not women, in the lowest bilirubin quartile (<0.6 mg/dl for men, <0.5 for women) had higher odds of prevalent diabetes compared to those in the highest quartile (>0.8 mg/dl for men, >0.6 mg/dl for women). We did not find any significant association between bilirubin and other CV risk factors in men or women.

Our finding of an inverse relationship between total bilirubin and diabetes is in line with the increasing literature on bilirubin's inverse association with cardiometabolic risk factors [6–13, 15–17]. In a cross-sectional analysis of the 1999–2006 U.S. National Health and Nutrition

**Table 1.** Baseline characteristics by bilirubin quartile in males in MASALA, 2010–2018.

| Exam 1: 477, Exam 1A: 121 | Overall | Quartile 1 (Male: ≤0.5 mg/dl) | Quartile 2 (Male: 0.6–0.7 mg/dl) | Quartile 3 (Male: 0.8, mg/dl) | Quartile 4 (Male: >0.8 mg/dl) |
|---|---|---|---|---|---|
| N | 598 | 177 | 131 | 161 | 129 |
| **Demographics** | | | | | |
| Age, mean (SD) | 58 (10) | 57 (9) | 58 (11) | 58 (9) | 57 (10) |
| ≥60 years | 258 (43%) | 74 (42%) | 62 (47%) | 71 (44%) | 51 (40%) |
| Education | | | | | |
| < Bachelor's Degree | 63 (11%) | 19 (11%) | 15 (11%) | 18 (11%) | 11 (9%) |
| Bachelor's Degree | 385 (64%) | 111 (62%) | 90 (69%) | 98 (61%) | 86 (67%) |
| >Bachelor's Degree | 150 (25%) | 47 (27%) | 26 (20%) | 45 (28%) | 32 (25%) |
| Income Category | | | | | |
| <$40,000 | 84 (14%) | 31 (18%) | 13 (10%) | 20 (12%) | 20 (16%) |
| $40–75,000 | 84 (14%) | 30 (17%) | 18 (14%) | 22 (14%) | 14 (11%) |
| $75,000–100,000 | 59 (10%) | 17 (10%) | 14 (11%) | 17 (11%) | 11 (9%) |
| >$100,000 | 371 (62%) | 99 (56%) | 86 (66%) | 102 (63%) | 84 (65%) |
| Percent Lived in U.S. | | | | | |
| <40% | 167 (28%) | 54 (31%) | 36 (28%) | 43 (27%) | 34 (26%) |
| 40–60% | 267 (45%) | 75 (42%) | 65 (50%) | 69 (43%) | 58 (45%) |
| >60% | 163 (27%) | 48 (27%) | 30 (24%) | 48 (30%) | 37 (29%) |
| **Sociobehavioral Factors** | | | | | |
| Smoking Category | | | | | |
| Never | 423 (71%) | 123 (70%) | 95 (73%) | 114 (71%) | 91 (71%) |
| Ever smoker | 175 (29%) | 54 (30%) | 36 (27%) | 47 (29%) | 38 (29%) |
| Alcohol Consumption | | | | | |
| Never Drinker (%) | 332 (56%) | 106 (60%) | 77 (59%) | 85 (53%) | 64 (50%) |
| Exercise, mean (SD) | 1416 (1413) | 1271 (1205) | 1466 (1336) | 1539 (1611) | 1410 (1487) |
| ≥600 MET-min/wk (%) | 319 (67%) | 83 (60%) | 76 (69%) | 78 (68%) | 82 (73%) |
| Dietary Consumption [N = 122 missing] | | | | | |
| Total caloric intake (kcal) | 1758 (570) | 1738 (580) | 1747 (570) | 1815 (559) | 1733 (570) |
| % kcal from fat | 29 (5) | 29 (5) | 29 (6) | 29 (4) | 28 (6) |
| % kcal from carbohydrates | 56 (6) | 56 (6) | 57 (7) | 56 (5) | 57 (7) |
| % kcal from protein | 14 (2) | 14 (2) | 15 (2) | 15 (2) | 14 (2) |
| **Physical Exam factors:** | | | | | |
| BMI (kg/m2) | 25.9 (3.7) | 26.4 (4.2) | 25.6 (3.3) | 26.0 (3.7) | 25.4 (3.1) |
| <22.9 | 130 (21%) | 35 (20%) | 30 (23%) | 34 (21%) | 31 (24%) |
| 23–27.4 | 296 (49%) | 84 (47%) | 68 (52%) | 80 (50%) | 64 (50%) |
| 27.5+ | 171 (29%) | 58 (33%) | 32 (25%) | 47 (29%) | 34 (26%) |
| Waist circumference | 96.7 (9.3) | 97.8 (10.5) | 95.6 (8.4) | 97.0 (9.0) | 95.5 (8.6) |
| Systolic BP | 127 (15) | 127 (14) | 126 (13) | 128 (14) | 129 (18) |
| Diastolic BP | 77 (9) | 76 (10) | 76 (9) | 78 (9) | 78 (10) |
| **Laboratory/Imaging Factors** | | | | | |
| HbA1c % | 6.14 (0.91) | 6.43 (1.12) | 6.05 (0.75) | 6.09 (0.81) | 5.88 (0.69) |
| HDL-c (mg/dl) | 44.8 (10.7) | 43.4 (10.0) | 44.2 (11.3) | 45.5 (10.6) | 46.0 (11.0) |
| <40 mg/dl for male | 200 (33%) | 68 (38%) | 51 (39%) | 45 (28%) | 36 (28%) |
| LDL-c (mg/dl) [N = 6 missing] | 108 (33) | 107 (32) | 107 (33) | 106 (33) | 112 (33) |
| >160 mg/dl (%) | 80 (13%) | 27 (16%) | 15 (11%) | 18 (11%) | 20 (16%) |
| Total Cholesterol (mg/dl) | 180 (38) | 179 (40) | 180 (39) | 178 (37) | 184 (36) |
| Triglycerides (mg/dl) | 139 (80) | 149 (90) | 143 (62) | 131 (62) | 137 (95) |

(*Continued*)

**Table 1.** (Continued)

| Exam 1: 477, Exam 1A: 121 | Overall | Quartile 1 (Male: ≤0.5 mg/dl) | Quartile 2 (Male: 0.6–0.7 mg/dl) | Quartile 3 (Male: 0.8, mg/dl) | Quartile 4 (Male: >0.8 mg/dl) |
|---|---|---|---|---|---|
| >150 mg/dl (%) | 206 (34%) | 68 (38%) | 52 (40%) | 43 (27%) | 43 (33%) |
| Total bilirubin (mg/dl) | 0.70 (0.26) | 0.45 (0.06) | 0.60 (0.0) | 0.74 (0.04) | 1.10 (0.22) |
| Coronary Artery Calcium (CAC) [N = 6 missing] | | | | | |
| CAC >0 (%) | 375 (63%) | 121 (69%) | 77 (60%) | 102 (64%) | 75 (59%) |
| 0 | 217 (37%) | 55 (31%) | 52 (40%) | 58 (36%) | 52 (41%) |
| 1–400 | 296 (51%) | 98 (56%) | 60 (47%) | 77 (48%) | 61 (48%) |
| >400 | 79 (13%) | 23 (13%) | 17 (13%) | 25 (16%) | 14 (11%) |
| Common carotid IMT, mm | 0.91 (0.24) | 0.91 (0.23) | 0.94 (0.31) | 0.92 (0.21) | 0.89 (0.25) |
| Internal carotid IMT, mm | 1.27 (0.49) | 1.31 (0.51) | 1.33 (0.60) | 1.17 (0.35) | 1.27 (0.48) |
| High Risk of 10-year ASCVD (> = 7.5%) [N = 7 missing] | 326 (55%) | 99 (58%) | 72 (55%) | 86 (54%) | 69 (54%) |
| HOMA-IR Score, median (IQR) | 2.79 (1.87–4.34) | 2.80 (1.98–4.94) | 2.82 (1.77–4.83) | 2.91 (2.02–4.28) | 2.72 (1.72–3.81) |
| **Comorbidities** | | | | | |
| Hypertension | 359 (60%) | 109 (62%) | 70 (53%) | 100 (62%) | 80 (62%) |
| Type 2 Diabetes | 173 (29%) | 71 (40%) | 31 (24%) | 46 (29%) | 25 (19%) |
| Dyslipidemia | 434 (73%) | 137 (77%) | 102 (78%) | 107 (66%) | 88 (68%) |
| Metabolic Syndrome | 232 (39%) | 74 (42%) | 54 (41%) | 63 (39%) | 41 (32%) |
| **Medication Use** | | | | | |
| Cholesterol-reducing medication Use | 220 (36%) | 64 (30%) | 51 (39%) | 63 (38%) | 42 (32%) |
| Statin medication use | 204 (34%) | 59 (33%) | 46 (35%) | 59 (36%) | 40 (31%) |
| Antihypertensive medication use | 229 (38%) | 73 (41%) | 37 (29%) | 65 (40%) | 54 (42%) |
| Insulin use | 11 (1.8%) | 4 (2.3%) | 2 (1.5%) | 4 (2.5%) | 1 (0.8%) |
| Metformin use | 118 (20%) | 50 (28%) | 18 (14%) | 34 (21%) | 16 (12%) |
| Non-Insulin diabetes medication use | 129 (22%) | 53 (30%) | 23 (18%) | 36 (23%) | 17 (13%) |

Format: For continuous variables, values are presented as mean (SD), unless otherwise specified. For categorical variables, values are presented as N (%). IMT = intima-media thickness.

Examination Survey (NHANES), Cheriyath et al. observed a higher prevalence of T2D in participants with total bilirubin <0.58 mg/dl compared to ≥0.58 mg/dl [31]. Several East Asian cohorts, including Korean [15–17, 20], Japanese [12], and Chinese [21, 32] cohorts, demonstrated similar findings. Among healthy, non-diabetic Japanese adults and Korean adults with diabetes, total bilirubin was inversely associated with HbA1c; among Chinese adults with impaired glucose tolerance, those in the lowest bilirubin quartile (<0.48 mg/dl) had significantly increased risk of incident T2D; and several longitudinal Korean studies have found increased risk of diabetes among patients with low bilirubin (<0.9 mg/dl). Cumulatively, a meta-analysis [14] demonstrated a 23% significant decrease in odds of prevalent diabetes when comparing the highest to lowest bilirubin tertile. Although there has been no previous literature on bilirubin's role among South Asians, our findings corroborate the aforementioned studies and suggest, that while the distribution of bilirubin may be different as evidenced by the various quartile cutoffs, the overall associations still persist.

Mendelian randomization studies aiming to determine whether bilirubin causes diabetes and CV risk have been inconsistent [10, 33, 34]. In a study by Abbasi et al. [34], among 8,592 Dutch participants, investigators found a causal association between a uridine diphosphate–glucuronosyltransferase locus, a gene responsible for conjugating and excreting bilirubin, and

**Table 2. Baseline characteristics by bilirubin quartile in females in MASALA, 2010–2018.**

| Exam 1: 415, Exam 1A: 136 | Overall | Quartile 1 (≤0.4 mg/dl) | Quartile 2 (0.5 mg/dl) | Quartile 3 (0.6 mg/dl) | Quartile 4 (>0.6 mg/dl) |
|---|---|---|---|---|---|
| N | 551 | 199 | 153 | 90 | 109 |
| **Demographics** | | | | | |
| Age, mean (SD) | 56 (9) | 55 (9) | 56 (9) | 57 (9) | 55 (9) |
| ≥60 years | 190 (34%) | 61 (31%) | 59 (39%) | 40 (44%) | 30 (28%) |
| Education | | | | | |
| < Bachelor's Degree | 94 (17%) | 34 (17%) | 24 (16%) | 15 (17%) | 21 (19%) |
| Bachelor's Degree | 259 (47%) | 102 (51%) | 68 (44%) | 41 (46%) | 48 (44%) |
| >Bachelor's Degree | 198 (36%) | 63 (32%) | 61 (40%) | 34 (38%) | 40 (37%) |
| Income Category | | | | | |
| <$40,000 | 91 (18%) | 32 (16%) | 25 (16%) | 21 (23%) | 19 (17%) |
| $40–75,000 | 69 (13%) | 21 (11%) | 23 (15%) | 11 (12%) | 14 (13%) |
| $75,000–100,000 | 61 (11%) | 15 (7.5%) | 26 (17%) | 9 (10%) | 11 (10%) |
| >$100,000 | 324 (59%) | 131 (66%) | 79 (52%) | 49 (54%) | 65 (60%) |
| Percent Lived in U.S. | | | | | |
| <40% | 166 (30%) | 58 (29%) | 48 (32%) | 20 (22%) | 40 (37%) |
| 40–60% | 235 (43%) | 91 (46%) | 67 (44%) | 38 (41%) | 39 (36%) |
| >60% | 149 (27%) | 50 (25%) | 38 (24%) | 32 (37%) | 29 (27%) |
| **Sociobehavioral Factors** | | | | | |
| Smoking Category | | | | | |
| Never | 536 (97%) | 191 (96%) | 149 (97%) | 89 (99%) | 107 (98%) |
| Ever smoker | 15 (3%) | 8 (4%) | 4 (3%) | 1 (1%) | 2 (2%) |
| Alcohol Consumption | | | | | |
| Never Drinker (%) | 454 (82%) | 165 (83%) | 125 (82%) | 75 (83%) | 89 (82%) |
| Exercise, mean (SD) | 1244 (1305) | 1135 (1182) | 1308 (1195) | 1217 (1259) | 1374 (1658) |
| ≥600 MET-min/wk (%) | 261 (63%) | 107 (63%) | 74 (69%) | 41 (65%) | 39 (53%) |
| Dietary Consumption (N = 136 missing) | | | | | |
| Total caloric intake (kcal) | 1571 (437) | 1552 (414) | 1589 (447) | 1598 (498) | 1567 (428) |
| % kcal from fat | 29.8 (4.7) | 29.9 (4.6) | 29.4 (5.3) | 29.1 (4.6) | 30.4 (4.7) |
| % kcal from carbohydrates | 56.2 (5.7) | 56.1 (5.2) | 56.7 (6.3) | 56.3 (5.3) | 55.7 (6.1) |
| % kcal from protein | 15.0 (2.2) | 15.0 (2.2) | 15.0 (2.2) | 15.5 (2.1) | 14.9 (2.2) |
| **Examination Factors** | | | | | |
| BMI (kg/m2) | 26.5 (4.4) | 26.6 (4.1) | 26.7 (4.7) | 26.2 (4.7) | 26.5 (4.2) |
| <22.9 | 119 (22%) | 34 (17%) | 33 (22%) | 25 (27%) | 27 (25%) |
| 23–27.4 | 231 (42%) | 92 (46%) | 62 (40%) | 32 (36%) | 45 (41%) |
| 27.5+ | 201 (36%) | 73 (37%) | 58 (38%) | 33 (37%) | 37 (34%) |
| Waist circumference | 90.5 (10.3) | 90.2 (9.5) | 91.0 (10.5) | 89.4 (10.8) | 91.2 (11.1) |
| Systolic BP | 124 (17) | 124 (16) | 124 (16) | 123 (18) | 124 (18) |
| Diastolic BP | 71 (10) | 70 (10) | 71 (10) | 71 (10) | 72 (10) |
| **Laboratory/Imaging Factors** | | | | | |
| HbA1c % | 5.96 (0.80) | 6.02 (0.80) | 6.04 (0.80) | 5.92 (1.10) | 5.85 (0.66) |
| HDL-c (mg/dl) | 56.2 (13.9) | 54.7 (14.4) | 56.1 (12.7) | 57.0 (14.5) | 57.3 (14.2) |
| <50 mg/dl (%) | 197 (36%) | 83 (42%) | 52 (34%) | 29 (32%) | 33 (30%) |
| LDL-c (mg/dl) | 114 (32) | 114 (30) | 117 (34) | 114 (32) | 108 (31) |
| >160 mg/dl (%) | 68 (12%) | 21 (11%) | 27 (18%) | 13 (14%) | 7 (6.4%) |
| Total Cholesterol (mg/dl) | 194 (36) | 193 (34) | 198 (40) | 195 (36) | 188 (34) |
| Triglycerides (mg/dl) | 121 (53) | 125 (56) | 123 (54) | 116 (47) | 115 (48) |

*(Continued)*

**Table 2.** (Continued)

| Exam 1: 415, Exam 1A: 136 | Overall | Quartile 1 (≤0.4 mg/dl) | Quartile 2 (0.5 mg/dl) | Quartile 3 (0.6 mg/dl) | Quartile 4 (>0.6 mg/dl) |
|---|---|---|---|---|---|
| >150 mg/dl (%) | 129 (23%) | 53 (27%) | 41 (27%) | 15 (17%) | 20 (18%) |
| Total bilirubin (mg/dl) | 0.54 (0.20) | 0.37 (0.05) | 0.50 (0.0) | 0.60 (0.0) | 0.85 (0.21) |
| Coronary Artery Calcium (CAC) [N = 2 missing] | | | | | |
| CAC >0 (%) | 162 (30%) | 60 (30%) | 38 (25%) | 31 (34%) | 33 (30%) |
| 0 | 387 (70%) | 139 (70%) | 113 (75%) | 59 (66%) | 76 (70%) |
| 1–400 | 143 (26%) | 56 (28%) | 30 (20%) | 28 (31%) | 29 (27%) |
| >400 | 19 (3.5%) | 4 (2.1%) | 8 (5.3%) | 3 (3.3%) | 4 (3.7%) |
| Common carotid IMT, mm | 0.84 (0.20) | 0.82 (0.17) | 0.87 (0.24) | 0.84 (0.20) | 0.82 (0.20) |
| Internal carotid IMT, mm | 1.14 (0.40) | 1.13 (0.31) | 1.16 (0.46) | 1.22 (0.50) | 1.07 (0.33) |
| High Risk of 10-year ASCVD (> = 7.5%) [N = 2 missing] | 97 (18%) | 32 (16%) | 31 (21%) | 18 (21%) | 16 (16%) |
| HOMA-IR Score, median (IQR) | 2.25 (1.48–3.36) | 2.30 (1.57–3.49) | 2.28 (1.52–3.33) | 1.82 (1.19–3.24) | 1.79 (1.21–2.98) |
| **Comorbidities** | | | | | |
| Hypertension | 253 (46%) | 98 (49%) | 66 (43%) | 42 (47%) | 47 (43%) |
| Type 2 Diabetes | 100 (18%) | 36 (18%) | 35 (23%) | 12 (13%) | 17 (15%) |
| Dyslipidemia | 339 (62%) | 129 (65%) | 98 (64%) | 52 (58%) | 60 (55%) |
| Metabolic Syndrome | 203 (37%) | 81 (41%) | 60 (39%) | 28 (31%) | 34 (31%) |
| **Medication Use** | | | | | |
| Cholesterol-reducing medication Use | 141 (26%) | 53 (26%) | 37 (24%) | 20 (22%) | 31 (28%) |
| Statin medication use | 132 (24%) | 46 (23%) | 36 (24%) | 19 (21%) | 31 (28%) |
| Antihypertensive medication use | 155 (25%) | 60 (30%) | 43 (28%) | 25 (28%) | 27 (25%) |
| Insulin use | 9 (1.6%) | 4 (2.0%) | 3 (2.0%) | 1 (1.1%) | 1 (0.9%) |
| Metformin use | 67 (12%) | 21 (11%) | 24 (16%) | 10 (11%) | 12 (11%) |
| Non-Insulin diabetes medication use | 77 (14%) | 27 (13%) | 26 (17%) | 11 (13%) | 13 (12%) |

**Table 3. Association between total bilirubin quantiles and type 2 diabetes in MASALA.**

| | Men | | | Women | | |
|---|---|---|---|---|---|---|
| | N | Unadjusted OR | Adjusted OR | N | Unadjusted OR | Adjusted OR |
| Quartile 1 | 71/177 | 2.79 (1.64, 4.73)* | 2.84 (1.60, 5.02)* | 36/199 | 1.20 (0.64, 2.24) | 1.21 (0.61, 2.41) |
| Quartile 2 | 31/131 | 1.29 (0.71, 2.34) | 1.41 (0.75, 2.65) | 35/146 | 1.61 (0.85, 3.05) | 1.62 (0.81, 3.27) |
| Quartile 3 | 46/161 | 1.66 (0.96, 2.90) | 1.42 (0.78, 2.57) | 12/86 | 0.83 (0.38, 1.85) | 0.82 (0.35, 1.95) |
| Quartile 4 | 25/129 | 1.00 (REF) | 1.00 (REF) | 17/104 | 1.00 (REF) | 1.00 (REF) |
| Tertile 1 | 71/177 | 2.17 (1.39, 3.39)* | 2.48 (1.52, 4.04)* | 9/56 | 1.12 (0.50, 2.54) | 1.30 (0.53, 3.20) |
| Tertile 2 | 56/226 | 1.07 (0.68, 1.67) | 1.23 (0.76, 1.99) | 62/296 | 1.55 (0.96, 2.52) | 1.60 (0.94, 2.72) |
| Tertile 3 | 46/195 | 1.00 (REF) | 1.00 (REF) | 29/199 | 1.00 (REF) | 1.00 (REF) |

Format: N: # of diabetes / total # of subjects. Odds Ratio [95% Confidence Interval].

*p-value of parameter estimate is <0.004 (Bonferroni corrected p-value threshold for significance)

Quartiles: Men: <0.6, 0.6, 0.7–0.8, >0.8; Women: <0.5, 0.5, 0.6, >0.6 mg/dl

Tertiles defined as follows: Male: <0.6, 0.6–0.7, >0.7; Female: <0.4, 0.4–0.5, >0.5 mg/dl.

Odds ratios adjusted for age, BMI, education level, household income quartile, percent of life living in the U.S., exercise (in MET-min/week), alcohol consumption (yes/no), metabolic syndrome criteria (except for outcome of interest: HbA1c%, waist circumference, HDL, triglycerides, systolic BP), LDL, liver fat attenuation, and smoking status (only in males).

Diabetes defined as fasting plasma glucose ≥126 mg/dl, 2-hour post-challenge glucose ≥200 mg/dL, HbA1c ≥6.5%, or anti-diabetes medication use.

**Table 4. Association between total bilirubin quartiles and type 2 diabetes among participants without liver dysfunction or CT-based fatty liver.**

| | Men | | | Women | | |
|---|---|---|---|---|---|---|
| | N | Unadjusted | Adjusted | N | Unadjusted | Adjusted |
| Quartile 1 | 67/169 | 2.24 (1.23, 4.08)* | 2.19 (1.15, 4.20) | 34/194 | 1.15 (0.58, 2.31) | 1.24 (0.57, 2.71) |
| Quartile 2 | 30/125 | 1.08 (0.55, 2.14) | 1.12 (0.54, 2.32) | 33/146 | 1.45 (0.71, 2.95) | 1.59 (0.71, 3.56) |
| Quartile 3 | 42/148 | 1.38 (0.72, 2.64) | 1.42 (0.78, 2.82) | 11/86 | 0.60 (0.24, 1.50) | 0.56 (0.20, 1.55) |
| Quartile 4 | 24/121 | 1.00 (REF) | 1.00 (REF) | 16/104 | 1.00 (REF) | 1.00 (REF) |

Format: N: # of diabetes / total # of subjects. Odds Ratio [95% Confidence Interval].

*p-value of parameter estimate is <0.004 (Bonferroni corrected p-value threshold for significance)

1. Odds ratios adjusted for age, BMI, education level, household income quartile, percent of life living in the U.S., exercise (in MET-min/week), alcohol consumption (yes/no), metabolic syndrome criteria (except for outcome of interest: HbA1c%, waist circumference, HDL, triglycerides, systolic BP), LDL, and smoking status (only in males).

Diabetes defined as fasting plasma glucose ≥126 mg/dl, 2-hour post-challenge glucose ≥200 mg/dL, HbA1c ≥6.5%, or anti-diabetes medication use.

T2D risk using a Mendelian randomization design. However, McArdle et al. [10], in a separate Mendelian randomization study of an Amish community in the U.S., utilized similar loci but found that while serum bilirubin was associated with CV risk factors, the loci were not. This study suggested that bilirubin was not causally linked to diabetes and may be mediated by an intermediary factor.

There are several plausible mechanisms for bilirubin's protective association with diabetes. In-vitro and in-vivo experimental studies have shown that unconjugated bilirubin can act as a potent, lipid-soluble antioxidant and help prevent oxidative stress, which in turn may reduce insulin resistance [11]. Correspondingly, patients with Gilbert Syndrome, a benign condition characterized by dysfunctional uridine-diphosphoglucuronate glucuronosyltransferase

**Table 5. Odds of cardiovascular risk factors in the lowest bilirubin vs. highest bilirubin quartile.**

| | Men | | Women | |
|---|---|---|---|---|
| | Unadjusted | Adjusted[1] | Unadjusted | Adjusted |
| Type 2 Diabetes | 2.79 (1.64, 4.73)* | 2.84 (1.60, 5.02)* | 1.20 (0.64, 2.24) | 1.21 (0.61, 2.41) |
| Hypertension | 0.98 (0.62, 1.57) | 0.60 (0.35, 1.04) | 1.28 (0.80, 2.05) | 1.23 (0.45, 1.42) |
| Dyslipidemia | 1.60 (0.96, 2.66) | 1.16 (0.67, 2.01) | 1.51 (0.94, 2.42) | 1.38 (0.82, 2.32) |
| Low HDL-cholesterol | 1.61 (1.00, 2.63) | 1.32 (0.76, 2.29) | 1.65 (1.00, 2.71) | 1.42 (0.81, 2.47) |
| Triglyceridemia | 1.25 (0.78, 2.01) | 0.99 (0.57, 1.73) | 1.62 (0.91, 2.88) | 1.38 (0.72, 2.64) |
| High LDL-cholesterol | 1.01 (0.54, 1.89) | 0.64 (0.31, 1.29) | 1.72 (0.71, 4.18) | 1.56 (0.62, 3.95) |
| Obesity | 1.36 (0.82, 2.25) | 1.35 (0.78, 2.34) | 1.13 (0.69, 1.84) | 1.00 (0.59, 1.70) |
| HOMA-IR ≥2[3] | 1.46 (0.83, 2.56) | 1.29 (0.64, 2.59) | 1.67 (0.93, 2.99) | 1.13 (0.58, 2.20) |
| Any Presence of CAC | 1.53 (0.95, 2.46) | 1.48 (0.82, 2.64) | 0.99 (0.60, 1.65) | 0.91 (0.50, 1.68) |
| High CAC[2] | 1.55 (0.72, 3.34) | 1.85 (0.71, 4.84) | 0.55 (0.13, 2.25) | 0.46 (0.10, 2.22) |
| High distal common carotid IMT | 1.18 (0.68, 2.07) | 1.10 (0.56, 2.16) | 1.07 (0.47, 2.46) | 0.88 (0.35, 2.22) |
| High internal carotid IMT | 1.01 (0.56, 1.82) | 0.98 (0.49, 1.98) | 1.86 (0.73, 4.77) | 1.79 (0.63, 5.05) |
| ASCVD Risk Score ≥7.5% | 0.86 (0.54, 1.37) | 1.11 (0.68, 1.81) | 0.91 (0.47, 1.74) | 1.18 (0.59, 2.34) |

Format: Odds Ratio [95% Confidence Interval]. IMT = intima-media thickness. CAC = coronary artery calcium.

*p-value of parameter estimate is <0.004 (Bonferroni corrected p-value threshold for significance)

1. Odds ratios adjusted for age, BMI, education level, household income quartile, percent of life living in the U.S., exercise (in MET-min/week), alcohol consumption (yes/no), metabolic syndrome criteria (except for outcome of interest: HbA1c%, waist circumference, HDL, triglycerides, systolic BP), LDL, and smoking status (only in males).

2. High CAC was a categorical variable with three categories (0, 1–399, ≥400). We estimated relative odds ratios using multinomial logistic regression.

**Table 6. Association between total bilirubin tertiles and incident type 2 diabetes by gender.**

| | Men (N = 294) | | Women (N = 279) | |
|---|---|---|---|---|
| | N | Unadjusted | N | Unadjusted |
| Tertile 1 | 5/77 | 0.47 (0.16, 1.38) | 4/36 | 2.97 (0.71, 12.82) |
| Tertile 2 | 10/116 | 0.64 (0.27, 1.53) | 13/144 | 2.36 (0.75, 7.45) |
| Tertile 3 | 13/101 | 1.00 (REF) | 4/99 | 1.00 (REF) |
| Fisher P-value | 0.38 | | 0.22 | |

Format: N: # of diabetes / total # of subjects. Odds Ratio [95% Confidence Interval].

*p-value of parameter estimate is <0.004 (Bonferroni corrected p-value threshold for significance). P-value calculated using Fisher's Exact test.

(UDP-GT) that prevents conjugation of unconjugated bilirubin, have exhibited decreased incidence of coronary artery disease and metabolic syndrome [10]. Bilirubin also has anti-inflammatory properties which can reduce plasma pro-inflammatory markers and inflammation-induced β-cell impairment. In mouse models, biliverdin (bilirubin precursor) administration prevented impaired glucose tolerance and reduced oxidative stress [35]. Bilirubin may also be a marker of basal liver function–higher levels within the normal range may be suggestive of healthier liver function and glucose metabolism, independent of ALT and AST. On the contrary, increases in bilirubin may be a reactive response to diabetes onset and occur due to increasing oxidative stress and subsequent stimulation of heme oxygenase-1, which then catalyzes formation of biliverdin and eventually, bilirubin [36].

We also observed that men had a significantly stronger association between bilirubin and diabetes than women. This may, in part, be due to lower prevalence of diabetes in South Asian women compared to men in MASALA. However, such sex differences have been described in the Framingham Offspring cohort and in NHANES [6, 36], suggesting a potential physiological mechanism. Testosterone has been shown to inhibit UDP- glucuronosyltransferase, decreasing bilirubin metabolism and potentially producing greater antioxidant and anti-inflammatory effects [37]. Sex hormones also affect body composition and body fat distribution, both of which can impact bilirubin metabolism as per adipose tissue-induced hormones. Paradoxically, we noticed that the direction of association for prevalent diabetes in men vs. women was opposite that of incident diabetes. If not an artifact of sample size, this might again point to the notion that bilirubin has less to do with causing diabetes but rather is a consequence of diabetes (e.g. released upon activation of heme oxygenase-1 in response to oxidative stress). Sex differences may also be the result of greater contribution of other traditional risk factors for diabetes onset in men vs. women.

In contrast to other studies, we did not find any significant associations between bilirubin and other CV risk factors such as hypertension, dyslipidemia, and atherosclerosis. Possible reasons include 1) unique pathological mechanisms for these conditions in South Asians independent of bilirubin and diabetes or unique genetic differences in expression of key bilirubin genes [38, 39]; 2) an older study population with multiple comorbidities that limit the role of bilirubin or that resulted in changes in bilirubin levels after disease onset; or 3) bias due to inclusion of participants taking statins, other lipid-lowering medications, or metformin which may interact with bilirubin metabolism. Our findings were unchanged when restricting patients with diabetes to only those using metformin. Metformin is a heme oxygenase-1 inhibitor [40], suggesting patients with diabetes using metformin may have lower bilirubin levels than patients not using metformin.

Interestingly, we found that the association between bilirubin and T2D was non-linear, with an increase in odds of diabetes in the 3rd quartile. This was again unique to our study

population, suggesting that we may not have adequately distinguished pathologic elevations in bilirubin from physiologically high bilirubin, and 4) limited power to clearly determine associations

Our study has several strengths, including a well-defined cohort of U.S. South Asians free from cardiovascular disease, adjustment for immigrant-specific factors such as number of years spent in the U.S., and radiographic measurement of subclinical atherosclerosis and fatty liver. Our study had several limitations as well, including 1) its cross-sectional nature, precluding any interpretation on temporality or causality; however, we did explore the association between baseline bilirubin level and incident diabetes but found no clear pattern; 2) low frequencies for certain covariate categories such as smoking as well as outcomes within bilirubin quartiles, which forced us to exclude smoking status from the analysis in women; the small number of cases may have led to the wide confidence intervals of our effect estimates in women. However, in sensitivity analyses, we found minimal change in the odds of diabetes after including these covariates, suggesting the overall conclusion would be identical, 3) lack of clinical data on hemoglobin levels, hepatitis serology, and other conditions that may affect bilirubin levels; 4) inability to distinguish unconjugated vs. conjugated bilirubin levels, which precluded us from making more mechanistic insights. However, the majority of observational studies have used total bilirubin, allowing for comparability; and 5) residual confounding from unmeasured or incompletely measured confounders (e.g. non-leisure-time physical activity).

In conclusion, among SA adults in the United States, we found an inverse association between total bilirubin and T2D but not other cardiometabolic factors. This association was stronger in men than in women. Longitudinal studies among SAs are needed to uncover total bilirubin's predictive or prognostic value in T2D and overall CV risk before it can be used clinically to improve CV risk stratification; nevertheless, our findings add to a growing number of studies in other populations purporting the role of bilirubin on CVD detection and prevention.

## Supporting information

**S1 File. The following supporting information (S1-S5 Tables) file includes supplemental data on subgroup, sensitivity, and exploratory analyses.**
(DOCX)

## Author Contributions

**Conceptualization:** Aayush Visaria, Alka Kanaya, Jaya Satagopan.

**Formal analysis:** Aayush Visaria, Jaya Satagopan.

**Funding acquisition:** Alka Kanaya.

**Investigation:** Aayush Visaria, Alka Kanaya, Jaya Satagopan.

**Methodology:** Soko Setoguchi, Meghana Gadgil, Jaya Satagopan.

**Project administration:** Alka Kanaya.

**Resources:** Alka Kanaya, Soko Setoguchi, Meghana Gadgil.

**Supervision:** Alka Kanaya, Soko Setoguchi, Meghana Gadgil, Jaya Satagopan.

**Validation:** Alka Kanaya.

**Visualization:** Aayush Visaria.

**Writing – original draft:** Aayush Visaria.

**Writing – review & editing:** Aayush Visaria, Alka Kanaya, Soko Setoguchi, Meghana Gadgil, Jaya Satagopan.

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
