## [Decision Letter · Decision Letter 0]

21 Feb 2023

PONE-D-23-00961Inverse Association between Total Bilirubin and Type 2 Diabetes in U.S. South AsiansPLOS ONE

Dear Dr. Visaria,

Thank you for submitting your manuscript to PLOS ONE. After careful consideration, we feel that it has merit but does not fully meet PLOS ONE’s publication criteria as it currently stands. Therefore, we invite you to submit a revised version of the manuscript that addresses the points raised during the review process.

We look forward to receiving your revised manuscript.

Kind regards,

Fredirick Lazaro mashili, MD, PhD

Academic Editor

PLOS ONE

Journal Requirements:

**Additional Editor Comments:**

The authors need to refer to PLOS ONE instructions on how to present and cite tables and figures. Each table should follow a body of text that cite that particular table. In addition, there is way too much information and tables. The authors need to selected data which is necessary to support their argument and present it clearly.

Reviewers' comments:

Reviewer's Responses to Questions

**Comments to the Author**

1. Is the manuscript technically sound, and do the data support the conclusions?

Reviewer #1: Partly

Reviewer #2: Yes

2. Has the statistical analysis been performed appropriately and rigorously? 

Reviewer #1: No

Reviewer #2: Yes

3. Have the authors made all data underlying the findings in their manuscript fully available?

Reviewer #1: No

Reviewer #2: No

4. Is the manuscript presented in an intelligible fashion and written in standard English?

Reviewer #1: Yes

Reviewer #2: Yes

5. Review Comments to the Author

Reviewer #1: This is an important topic. however you have alot of data and i am sorry to say that you have overpresented the data. Please choose wisely your outcome and predictor variables, a few that will answer you objectives specifically for this paper. Your methods and results as well as discussion should focus on the research question that is being answered in this paper.

Reviewer #2: Abstract:

-Lacks introduction/background: please provide

-aims; did you knew the association between bilirubin existed prior to this study? it seems yours aim was just to test the direction of the association that existed. please state your aim clear and concisely

Introduction:

-Not adequately written

-Do you think your introduction is enough especially for new readers? It would be better if the areas under study are more described

-the similar study was done in Chinese, Japanese etc, why do the same study in SALA? did you think they differed?

Objectives:

-You state aim in the abstract but not in main work

Methods:

-this section is not adequately writen

-Study population: authors explains the exclusion and inclusion criteria instead of describing the nature of study participant

-Variables; which were your dependent and independent variables

-Ethical consideration: isn't there any ethical considerations to mention?

Result:

-what was the use of covariates? you have not reported anything in that case.

Discussion:

-Why do think you obtained similar results as Chinese and Japanese cohorts? why your results were not consistent with mendelian studies?

-What would you recommend based on your findings?

-

6. PLOS authors have the option to publish the peer review history of their article (what does this mean?). If published, this will include your full peer review and any attached files.

Reviewer #1: No

Reviewer #2: No

---

## [Author Response · Author response to Decision Letter 0]

31 Aug 2023

Responses to Reviewer #1:

1. This is an important topic. however you have alot of data and i am sorry to say that you have overpresented the data. Please choose wisely your outcome and predictor variables, a few that will answer you objectives specifically for this paper. Your methods and results as well as discussion should focus on the research question that is being answered in this paper.

Thank you for this important comment. We have now edited the manuscript and tables/figures to focus specifically on the association between total bilirubin and prevalent & incident type 2 diabetes. The other data presented on additional cardiovascular risk factors is now moved to the supplemental material as exploratory analyses.

Responses to Reviewer #2: 

Abstract:

-Lacks introduction/background: please provide

-aims; did you knew the association between bilirubin existed prior to this study? it seems yours aim was just to test the direction of the association that existed. please state your aim clear and concisely

Thank you for this comment. We have now revised the abstract to better introduce the topic and aims of the study. Several observational studies in the past have found an association between bilirubin and T2D, but the direction of the association and whether the association exists in South Asians is still unknown.

“Aims: United States South Asians constitute a fast-growing ethnic group with high prevalence of type 2 diabetes (T2D) despite lower mean BMI and other traditional risk factors compared to other races/ethnicities. Bilirubin has gained attention as a potential antioxidant, cardio-protective marker. Hence we sought to determine whether total bilirubin was associated with prevalent and incident T2D in U.S. South Asians.” (Abstract, Page 2)

Introduction:

-Not adequately written

-Do you think your introduction is enough especially for new readers? It would be better if the areas under study are more described

Thank you for this feedback. We have now revised the introduction to provide more detail into the reasons for the study and associated background material.

“South Asians (SA), individuals with origins in Bangladesh, Bhutan, India, the Maldives, Nepal, Pakistan, and Sri Lanka, in the U.S. have particularly high prevalence of T2D [2] and risk of downstream cardiovascular disease (CVD) despite lower mean BMI and prevalence of other traditional risk factors [3]. This increased risk of T2D and CVD is likely multifactorial, although causes are not well understood and are understudied.” (Introduction, Page 3, Line 77-81)

“In recent years, serum bilirubin has arisen as a potential protective factor for T2D and related CV factors. Traditionally, bilirubin, a byproduct of hemoglobin degradation, was known as a cytotoxic waste product in liver disease and hemolytic anemias [4,5]. However, stemming from observations of low prevalence of cardiometabolic conditions in patients with Gilbert’s Syndrome, a benign, genetic disorder characterized by elevated unconjugated bilirubin, bilirubin has gained attention as an endogenous antioxidant, cardio-protective marker. Several observational studies have since demonstrated an inverse association between total bilirubin and cardiovascular disease [6-9] & associated risk factors such as T2D, insulin resistance and dyslipidemia [10-14].” (Introduction, Page 3, Line 93-95)

-the similar study was done in Chinese, Japanese etc, why do the same study in SALA? did you think they differed?

Thank you for this insightful question. We wanted to determine the association between bilirubin and T2D in this previously unstudied high-risk ethnic group because South Asians have considerably higher prevalence of T2D than other races/ethnicities. The factors currently known to be associated with this high prevalence of T2D include higher prevalence of elevated visceral adiposity despite similar/lower BMIs, and more unfavorable lipid compositions. Both these factors largely affect or are driven by the liver; hence, we thought it would be important to understand whether bilirubin, which is also produced in the liver, is associated with T2D.

“Given the high prevalence of unfavorable lipid profiles and visceral adiposity in U.S. SAs, both of which can affect or are driven by the liver, we hypothesized that bilirubin, also produced in the liver, may be associated with T2D in SAs.” (Introduction, Page 3, Line 96-97)

Objectives:

-You state aim in the abstract but not in main work

Thank you for this comment. We have now made the objective more clear at the end of the introduction section.

“Hence, the objective of this study was to determine whether total bilirubin is independently, associated with both prevalent and incident T2D among a cohort of US South Asians.” (Introduction, Page 3, Line

Methods:

-this section is not adequately writen

-Study population: authors explains the exclusion and inclusion criteria instead of describing the nature of study participant

Thank you for this comment. We describe the study population in the Results section of the manuscript. In this Methods section 2.2 of the manuscript, we describe how we derived the study population from the original MASALA cohort that was described in Section 2.1. We hope this helps clarify the structure.

-Variables; which were your dependent and independent variables

Thank you for this comment. Our main independent variable was total bilirubin as described in Methods subsection 2.3. Our main dependent variable was prevalent T2D as described in Methods subsection 2.4.

-Ethical consideration: isn't there any ethical considerations to mention?

Thank you for this comment. We have now included sentence regarding ethical considerations. As this is a secondary data analysis of de-identified data, our study was deemed exempt per Rutgers IRB.

“This study was approved by University of California San Francisco’s Institutional Review Board and Rutgers’s Institutional Review Board.” (Methods, Section 2.1, Line 110-112)

Result:

-what was the use of covariates? you have not reported anything in that case.

Thank you for this comment. Covariates were used as potential confounders in the regression analysis in order to try and isolate the independent relationship between bilirubin and T2D. The statistical analysis section of the manuscript describes the statistical procedure used to account for potential confounders.

“We adjusted for potential confounders selected a priori, including age, BMI, exercise, alcohol consumption, education, income quartile, percent time in the U.S., and metabolic syndrome criteria (hemoglobin A1c%, systolic BP, HDL, triglycerides, except for outcome of interest), and LDL-cholesterol. We additionally adjusted for smoking status in males but not females due to a very low sample of female smokers.“ (Methods, Section 2.6, Lines 163-167)

Discussion:

-Why do think you obtained similar results as Chinese and Japanese cohorts? why your results were not consistent with mendelian studies?

Thank you for this question. I think the similar findings across various Asian ethnicities suggests that there is in fact a true association between bilirubin and T2D that is irrespective of culture or race/ethnicity but rather more physiologic in nature. Some of the speculative mechanisms driving this inverse association are described in the discussion (Lines 251-264). In terms of the Mendelian randomization studies, our results are consistent with several of them but what we meant to say in the discussion is that the Mendelian studies are not consistent with each other. In general, it is difficult to compare Mendelian studies to our study because of the vast differences in study methodology used. However, all studies do seem to suggest an association that may or may not be mediated by other unmeasured factors.

-What would you recommend based on your findings?

Thank you for this question. Further studies are needed before any clinical implications can be derived. I would recommend researchers further study bilirubin, including its unconjugated form, longitudinally in South Asians so that we can better establish whether T2D results in decreases in bilirubin or whether bilirubin can be used as a predictive marker.

Responses to Editor/Reviewer #3:

Minor.

1. This works lacks line numbers, making it difficult to refer

Thank you for this suggestion. We have now included line numbers.

2. In the covariates, what do metabolic factors mean?

Thank you for this question. To clarify this further, we changed the text from ‘metabolic factors’ to cardiovascular risk factors.

“…and CV risk factors (HDL-cholesterol, LDL-cholesterol, triglycerides, hypertension, waist circumference, and BMI-based obesity).” (Methods, Covariates, Page 5, Line 157-158)

3. What does total exercise represent? Physical activity? How about other activities that are not exercises?

Thank you for this question. Total exercise represents leisure-time physical activity only and does not include occupational or other physical activity. This is a limitation of our study as other forms of physical activity were not available for this study. We have now changed ‘total exercise’ to ‘leisure-time physical activity’ and included a sentence about the limitations in the Discussion.

“and 5) residual confounding from unmeasured or incompletely measured confounders (e.g. non-leisure-time physical activity).” (Discussion, Page 8, Line 305-306)

4. The primary outcome is not clear, both in the method and results section. Is it T2D or bilirubin?

Thank you for this comment. The primary outcome is prevalent T2D. We have made this clearer in the Methods section. Total bilirubin is our main independent variable.

“The primary outcome was T2D, defined by laboratory criteria (fasting plasma glucose ≥126 mg/dl or 2-hour post-challenge glucose ≥200 mg/dL), or use of any diabetes medications. The same definitions were used for both prevalent and incident T2D.” (Methods, Page 4, Line 125-127)

5. Results section 3.3 can be best presented before

Thank you for this suggestion. We have now restructured the results section to focus on the primary outcomes, including Section 3.3 before describing secondary outcomes. Section 3.3 is now Section 3.1.

Major

1. Authors aimed to investigate the inverse association and not just association. With the few studies done in this area, which some have reported inverse association and others have reported no association; authors could investigate whether there is any association in the target population regardless of the direction of the association.

Thank you for this comment. We apologize for the confusion – although we hypothesized there would be an inverse association between total bilirubin and T2D, we performed two-sided statistical tests meaning that we did not investigate the inverse association only but rather the general association regardless of direction. We have now clarified this in the text by removing mention of inverse association in our methods and specifying two-sided statistical tests.

“All analysis was done with two-sided statistical tests, using SAS 9.4 (Cary, NC) with an alpha level of 0.05.” (Methods, Statistical Analysis, Page 5, Line 183)

2. Authors have not shown clearly how they defined the outcome of interest, T2D. Show it clearly the results by FBG, 2-hr post OGTT or by HbA1c. The associations may be different and the explanations can differ too. In the current analysis, it is too general and it is difficult to understand the pathophysiology.

Thank you for this comment. Because this was a secondary data analysis of previously collected cohort data from the MASALA study, we did not have the ability to parse out the definitions used to define diabetes. As a result, we used a comprehensive clinical definition of diabetes to sensitively capture all persons with either fasting plasma glucose ≥126 mg/dl or 2-hour post-challenge glucose ≥200 mg/dL or use of any anti-diabetes medication. HbA1c was not used in our definition because of concerns that HbA1c may not be an accurate marker for diabetes diagnosis in South Asians, given its variability in patients with anemia and hemoglobinopathies. Even though we were not able to parse out the individual constituents of the criteria used to define diabetes, we believe the underlying pathophysiology across all these are similar and clinically are treated similarly.

“The primary outcome was T2D, defined by laboratory criteria (fasting plasma glucose ≥126 mg/dl or 2-hour post-challenge glucose ≥200 mg/dL), or use of any diabetes medications. The same definitions were used for both prevalent and incident T2D.” (Methods, Section 2.4, Page 4, Line 125-127)

3. Please define metabolic syndrome clearly, and state which criteria has been used in your definition.

Thank you for this comment. In an effort to condense our findings and focus on diabetes as the outcome, we have removed several of our secondary outcomes including metabolic syndrome. Metabolic syndrome criteria, as continuous measures, per the NCEP ATP III were still adjusted for in our regression analysis between bilirubin and diabetes as they may confound the association.

“We adjusted for potential confounders selected a priori, including age, BMI, exercise, alcohol consumption, education, income quartile, percent time in the U.S., and metabolic syndrome criteria per NCEP’s Adult Treatment Panel (ATP) III (hemoglobin A1c%, systolic BP, HDL, triglycerides, except for outcome of interest), and LDL-cholesterol” (Methods, Statistical Analysis, Page 5, Line 163-166)

4. Revise your materials and methods section. Clearly show how data for the outcome and predictor variables were collected and analyzed.

Thank you for this comment. Because this was a secondary data analysis, we did not go into detail into the data collection methodologies and laboratory methods as previous papers (cited in the Methods – reference 22) have explained it in great detail. Nevertheless, Section 2.3 explains how our main predictor variable (total bilirubin) was measured and categorized into sex-specific quartiles. Section 2.4 defines the primary outcome and secondary outcomes. We edited this section to be clearer about the primary outcome.

5. Why did authors use linear regression analysis verifying approximate normality of continuous outcomes? Please use the appropriate methods to check for normality and decide what tests of association will suite your data.

Thank you for this clarifying question. We verified normality using Q-Q plots and associated statistical tests which is customary prior to conducting linear regression analysis. We have presented both linear regression (for continuous secondary outcomes) and logistic regression (for yes/no secondary outcomes) because both provide useful information that is complementary to each other. Logistic regression analyses where the outcomes are binary are more clinical in nature and provide useful information about odds of clinical conditions. From a biological perspective, however, categorizing otherwise continuous variables may lose information about the continuous nature of an association. Hence, to demonstrate this as well we have presented linear regression analyses. All the secondary outcome analyses are meant to be exploratory so we have restructured the manuscript tables/figures so that all the main text tables are about diabetes, our primary outcome.

6. Authors used both quartiles and tertiles in their analyses. It is not clear why they didn’t choose either of the two in participants’ characteristics and logistic regression analysis?

Thank you for this question. Bilirubin quartiles were our primary classification strategy to define bilirubin. Hence, the participant characteristics are presented by bilirubin quartile. For the logistic regression analysis, we also looked at tertiles to determine the sensitivity of our findings – aka whether our findings were specific to the way we classified bilirubin rather than a true association.

7. What is the clear role of sensitivity analyses in this observational study?

Thank for you this insightful question. Sensitivity analyses are analyses done to ensure the robustness of the primary findings. By changing specific aspects of the bilirubin classification scheme (e.g. tertiles instead of quartiles) we can assess whether the primary findings are sensitive to the way bilirubin is classified and thus assess the associations regardless of how the exposure/outcome is classified or measured.

8. Authors have concluded that, they have found an inverse association between total bilirubin and T2D, but this was not the case in women, so it should be clearly stated.

Thank you for this comment. We have now made it clear in the title as well as the abstract conclusion that this association was present in men, not women.

---

## [Decision Letter · Decision Letter 1]

24 Oct 2023

PONE-D-23-00961R1Inverse Association between Total Bilirubin and Type 2 Diabetes in U.S. South Asian  Males but not FemalesPLOS ONE

Dear Dr. Visaria,

Thank you for submitting your manuscript to PLOS ONE. After careful consideration, we feel that it has merit but does not fully meet PLOS ONE’s publication criteria as it currently stands. Therefore, we invite you to submit a revised version of the manuscript that addresses the points raised during the review process.

ACADEMIC EDITOR: Please address all the comments given by the reviewer #3This manuscript requires a minor revisionPlease ensure that all the formatting and other requirements are according to the journals guidelines ==============================

We look forward to receiving your revised manuscript.

Kind regards,

Fredirick Lazaro mashili, MD, PhD

Academic Editor

PLOS ONE

Journal Requirements:

Additional Editor Comments:

You have adequately addressed the reviewers' comments. However, there are a few minor corrections that need to be made to ensure the manuscript is ready for publication.

Reviewers' comments:

Reviewer's Responses to Questions

**Comments to the Author**

1. If the authors have adequately addressed your comments raised in a previous round of review and you feel that this manuscript is now acceptable for publication, you may indicate that here to bypass the “Comments to the Author” section, enter your conflict of interest statement in the “Confidential to Editor” section, and submit your "Accept" recommendation.

Reviewer #3: (No Response)

Reviewer #4: All comments have been addressed

2. Is the manuscript technically sound, and do the data support the conclusions?

Reviewer #3: Yes

Reviewer #4: Yes

3. Has the statistical analysis been performed appropriately and rigorously? 

Reviewer #3: Yes

Reviewer #4: Yes

4. Have the authors made all data underlying the findings in their manuscript fully available?

Reviewer #3: No

Reviewer #4: Yes

5. Is the manuscript presented in an intelligible fashion and written in standard English?

Reviewer #3: Yes

Reviewer #4: Yes

6. Review Comments to the Author

Reviewer #3: I suggest the following minor corrections

1. Lines 99 - 101 "Given the high prevalence of unfavorable lipid profiles and visceral adiposity in U.S. SAs, both of which can affect or are driven by the liver, we hypothesized that bilirubin, also produced in the liver, may be associated with T2D in SAs. This assertion needs to be supported by reference.

2. Line 266 and 268 "Invitro and in vitro experimental studies have shown that unconjugated bilirubin can act as a potent, lipid-soluble antioxidant and help prevent oxidative stress that can induce insulin resistance". This statement is not clear.

3. Line 283-287. The description of the role of testosterone in bilirubin metabolism also needs to be supported by the relevant citation.

4. To enhance the germane of the study. I suggest, a description of the clinical relevance of the study.

Reviewer #4: Please counter check the study/analysis design on the methodology section of the abstract and the main manuscript. On the abstract it is written prospective analysis while on the main manuscript it says retrospective analysis. Please clarify and rectify that.

7. PLOS authors have the option to publish the peer review history of their article (what does this mean?). If published, this will include your full peer review and any attached files.

Reviewer #3: No

Reviewer #4: **Yes: **Fredirick Mashili

---

## [Author Response · Author response to Decision Letter 1]

1 Jan 2024

Dear Dr. Fredirick Lazaro Mashili,

Thank you for the additional comments and suggestions regarding our study (ID: PONE-D-23-00961_R1) titled, “Inverse Association between Total Bilirubin and Type 2 Diabetes in U.S. South Asian Males but not Females”. We have addressed all concerns and believe the comments have improved our manuscript. Below are the corrections/responses to each of the remarks. Changes to the manuscript text are also marked as tracked changes in the manuscript file.

Responses to Reviewer #3

1. Lines 99 - 101 "Given the high prevalence of unfavorable lipid profiles and visceral adiposity in U.S. SAs, both of which can affect or are driven by the liver, we hypothesized that bilirubin, also produced in the liver, may be associated with T2D in SAs. This assertion needs to be supported by reference.

Thank you for this comment. We have now cited this statement (citation #22).

Revision: “Given the high prevalence of unfavorable lipid profiles and visceral adiposity in U.S. SAs, both of which can affect or are driven by the liver [22], we hypothesized that bilirubin, also produced in the liver, may be associated with T2D in SAs.” (Page 3, Lines 93-95)

2. Line 266 and 268 "Invitro and in vitro experimental studies have shown that unconjugated bilirubin can act as a potent, lipid-soluble antioxidant and help prevent oxidative stress that can induce insulin resistance". This statement is not clear.

Thank you for this clarification question. We have now reworded the sentence to clarify it and correct some typographical mistakes.

Revision: “There are several plausible mechanisms for bilirubin’s protective association with diabetes. In-vitro and in-vivo experimental studies have shown that unconjugated bilirubin can act as a potent, lipid-soluble antioxidant and help prevent oxidative stress, which in turn may reduce insulin resistance [11].” (Page 7, Lines 256-258)

3. Line 283-287. The description of the role of testosterone in bilirubin metabolism also needs to be supported by the relevant citation.

Thank you for this comment. We have now cited this statement with a seminal in-vitro study looking at testosterone’s role in bilirubin metabolism.

Revision: “Testosterone has been shown to inhibit UDP- glucuronosyltransferase, decreasing bilirubin metabolism and potentially producing greater antioxidant and anti-inflammatory effects [37].” (Page 7, Lines 274-275)

4. To enhance the germane of the study. I suggest, a description of the clinical relevance of the study.

Thank you for this comment. We have now added 2 sentences to put our findings into clinical context.

Revision: “In conclusion, among SA adults in the United States, we found an inverse association between total bilirubin and T2D but not other cardiometabolic factors. This association was stronger in men than in women. Longitudinal studies among SAs are needed to uncover total bilirubin’s predictive or prognostic value in T2D and overall CV risk before it can be used clinically to improve CV risk stratification; nevertheless, our findings add to a growing number of studies in other populations purporting the role of bilirubin on CVD detection and prevention.” (Page 8, Lines 314-317)

Reviewer #4: Please counter check the study/analysis design on the methodology section of the abstract and the main manuscript. On the abstract it is written prospective analysis while on the main manuscript it says retrospective analysis. Please clarify and rectify that.

Thank you for this comment. We conducted both a cross-sectional (retrospective) and prospective, exploratory analysis. Our main analysis is retrospective in nature but because we also had prospective data on our outcome of interest, we also explored the association between bilirubin and incident diabetes. This was designated to be exploratory analysis only due to the limited sample size. We removed the reference to ‘retrospective study’ in the Methods section and changed it to ‘secondary analysis’ to avoid confusion.

Revision: “This was a secondary analysis of fully de-identified data.” (Methods, Page 3, Line 112)

Sincerely,

Aayush Visaria

---

## [Decision Letter · Decision Letter 2]

11 Jan 2024

Inverse Association between Total Bilirubin and Type 2 Diabetes in U.S. South Asian  Males but not Females

PONE-D-23-00961R2

Dear Dr. Visaria,

We’re pleased to inform you that your manuscript has been judged scientifically suitable for publication and will be formally accepted for publication once it meets all outstanding technical requirements.

Kind regards,

Fredirick Lazaro mashili, MD, PhD

Academic Editor

PLOS ONE

Additional Editor Comments (optional):

All the comments have been sufficiently addressed

Reviewers' comments:

Reviewer's Responses to Questions

**Comments to the Author**

1. If the authors have adequately addressed your comments raised in a previous round of review and you feel that this manuscript is now acceptable for publication, you may indicate that here to bypass the “Comments to the Author” section, enter your conflict of interest statement in the “Confidential to Editor” section, and submit your "Accept" recommendation.

Reviewer #3: All comments have been addressed

Reviewer #4: All comments have been addressed

2. Is the manuscript technically sound, and do the data support the conclusions?

Reviewer #3: Yes

Reviewer #4: Yes

3. Has the statistical analysis been performed appropriately and rigorously? 

Reviewer #3: Yes

Reviewer #4: I Don't Know

4. Have the authors made all data underlying the findings in their manuscript fully available?

Reviewer #3: Yes

Reviewer #4: Yes

5. Is the manuscript presented in an intelligible fashion and written in standard English?

Reviewer #3: (No Response)

Reviewer #4: Yes

6. Review Comments to the Author

Reviewer #3: Authors have addressed previous comments in a satisfactory manner, I would recommend accepting this work for publication.

Reviewer #4: All the comments raised by both the reviewers have been thoroughly and sufficiently addressed. The authors have provided all the missing citations as recommended

7. PLOS authors have the option to publish the peer review history of their article (what does this mean?). If published, this will include your full peer review and any attached files.

Reviewer #3: **Yes: **Oscar Mbembela

Reviewer #4: **Yes: **Fredirick mashili

---

## [Editor Report · Acceptance letter]

25 Jan 2024

PONE-D-23-00961R2 

PLOS ONE

Dear Dr. Visaria, 

I'm pleased to inform you that your manuscript has been deemed suitable for publication in PLOS ONE. Congratulations! Your manuscript is now being handed over to our production team.

Kind regards, 

on behalf of

Dr Fredirick Lazaro mashili 

Academic Editor

PLOS ONE